# Active Exploration for Learning Symbolic Representations

**Garrett Andersen**
PROWLER.io
Cambridge, United Kingdom
garrett@prowler.io

**George Konidaris**
Department of Computer Science
Brown University
gdk@cs.brown.edu

## Abstract

We introduce an online active exploration algorithm for data-efficiently learning an abstract symbolic model of an environment. Our algorithm is divided into two parts: the first part quickly generates an intermediate Bayesian symbolic model from the data that the agent has collected so far, which the agent can then use along with the second part to guide its future exploration towards regions of the state space that the model is uncertain about. We show that our algorithm outperforms random and greedy exploration policies on two different computer game domains. The first domain is an Asteroids-inspired game with complex dynamics but basic logical structure. The second is the Treasure Game, with simpler dynamics but more complex logical structure.

## 1 Introduction

Much work has been done in artificial intelligence and robotics on how high-level state abstractions can be used to significantly improve planning [19]. However, building these abstractions is difficult, and consequently, they are typically hand-crafted [15, 13, 7, 4, 5, 6, 20, 9].

A major open question is then the problem of abstraction: how can an intelligent agent learn high-level models that can be used to improve decision making, using only noisy observations from its high-dimensional sensor and actuation spaces? Recent work [11, 12] has shown how to automatically generate symbolic representations suitable for planning in high-dimensional, continuous domains. This work is based on the hierarchical reinforcement learning framework [1], where the agent has access to high-level skills that abstract away the low-level details of control. The agent then learns representations for the (potentially abstract) effect of using these skills. For instance, opening a door is a high-level skill, while knowing that opening a door typically allows one to enter a building would be part of the representation for this skill. The key result of that work was that the symbols required to determine the probability of a plan succeeding are directly determined by characteristics of the skills available to an agent. The agent can learn these symbols autonomously by exploring the environment, which removes the need to hand-design symbolic representations of the world.

It is therefore possible to learn the symbols by naively collecting samples from the environment, for example by random exploration. However, in an online setting the agent shall be able to use its previously collected data to compute an exploration policy which leads to better data efficiency. We introduce such an algorithm, which is divided into two parts: the first part quickly generates an intermediate Bayesian symbolic model from the data that the agent has collected so far, while the second part uses the model plus Monte-Carlo tree search to guide the agent's future exploration towards regions of the state space that the model is uncertain about. We show that our algorithm is significantly more data-efficient than more naive methods in two different computer game domains. The first domain is an Asteroids-inspired game with complex dynamics but basic logical structure. The second is the Treasure Game, with simpler dynamics but more complex logical structure.

## 2 Background

As a motivating example, imagine deciding the route you are going to take to the grocery store; instead of planning over the various sequences of muscle contractions that you would use to complete the trip, you would consider a small number of high-level alternatives such as whether to take one route or another. You also would avoid considering how your exact low-level state affected your decision making, and instead use an abstract (symbolic) representation of your state with components such as whether you are at home or an work, whether you have to get gas, whether there is traffic, etc. This simplification reduces computational complexity, and allows for increased generalization over past experiences. In the following sections, we introduce the frameworks that we use to represent the agent's high-level skills, and symbolic models for those skills.

### 2.1 Semi-Markov Decision Processes

We assume that the agent's environment can be described by a semi-Markov decision process (SMDP), given by a tuple $D = (S, O, R, P, \gamma)$, where $S \subseteq \mathbb{R}^d$ is a $d$-dimensional continuous state space, $O(s)$ returns a set of temporally extended actions, or *options* [19] available in state $s \in S$, $R(s', t, s, o)$ and $P(s', t \mid s, o)$ are the reward received and probability of termination in state $s' \in S$ after $t$ time steps following the execution of option $o \in O(s)$ in state $s \in S$, and $\gamma \in (0, 1]$ is a discount factor. In this paper, we are not concerned with the time taken to execute $o$, so we use $P(s' \mid s, o) = \int P(s', t \mid s, o) \mathrm{d}t$.

An option $o$ is given by three components: $\pi_o$, the *option policy* that is executed when the option is invoked, $I_o$, the *initiation set* consisting of the states where the option can be executed from, and $\beta_o(s) \to [0, 1]$, the *termination condition*, which returns the probability that the option will terminate upon reaching state $s$. Learning models for the initiation set, rewards, and transitions for each option, allows the agent to reason about the effect of its actions in the environment. To learn these *option models*, the agent has the ability to collect observations of the forms $(s, O(s))$ when entering a state $s$ and $(s, o, s', r, t)$ upon executing option $o$ from $s$.

### 2.2 Abstract Representations for Planning

We are specifically interested in learning option models which allow the agent to easily evaluate the success probability of plans. A *plan* is a sequence of options to be executed from some starting state, and it *succeeds* if and only if it is able to be run to completion (regardless of the reward). Thus, a plan $\{o_1, o_2, ..., o_n\}$ with starting state $s$ succeeds if and only if $s \in I_{o_1}$ and the termination state of each option (except for the last) lies in the initiation set of the following option, i.e. $s' \sim P(s' \mid s, o_1) \in I_{o_2}$, $s'' \sim P(s'' \mid s', o_2) \in I_{o_3}$, and so on.

Recent work [11, 12] has shown how to automatically generate a symbolic representation that supports such queries, and is therefore suitable for planning. This work is based on the idea of a *probabilistic symbol*, a compact representation of a distribution over infinitely many continuous, low-level states. For example, a probabilistic symbol could be used to classify whether or not the agent is currently in front of a door, or one could be used to represent the state that the agent would find itself in after executing its 'open the door' option. In both cases, using probabilistic symbols also allows the agent to be uncertain about its state.

The following two probabilistic symbols are provably sufficient for evaluating the success probability of any plan [12]; *the probabilistic precondition*: $\mathrm{Pre}(o) = P(s \in I_o)$, which expresses the probability that an option $o$ can be executed from each state $s \in S$, and *the probabilistic image operator*:

$$\mathrm{Im}(o, Z) = \frac{\int_S P(s' \mid s, o) Z(s) P(I_o \mid s) \mathrm{d}s}{\int_S Z(s) P(I_o \mid s) \mathrm{d}s},$$

which represents the distribution over termination states if an option $o$ is executed from a distribution over starting states $Z$. These symbols can be used to compute the probability that each successive option in the plan can be executed, and these probabilities can then be multiplied to compute the overall success probability of the plan; see Figure 1 for a visual demonstration of a plan of length 2.

**Subgoal Options** Unfortunately, it is difficult to model $\mathrm{Im}(o, Z)$ for arbitrary options, so we focus on restricted types of options. A *subgoal option* [17] is a special class of option where the distribution over termination states (referred to as the subgoal) is independent of the distribution over starting

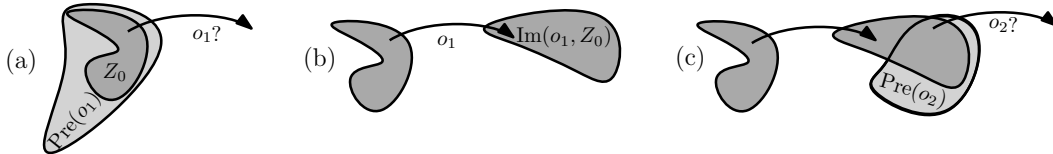

Figure 1: Determining the probability that a plan consisting of two options can be executed from a starting distribution $Z_0$. (a): $Z_0$ is contained in $\mathrm{Pre}(o_1)$, so $o_1$ can definitely be executed. (b): Executing $o_1$ from $Z_0$ leads to distribution over states $\mathrm{Im}(o_1, Z_0)$. (c): $\mathrm{Im}(o_1, Z_0)$ is not completely contained in $\mathrm{Pre}(o_2)$, so the probability of being able to execute $o_2$ is less than 1. Note that $\mathrm{Pre}$ is a set and $\mathrm{Im}$ is a distribution, and the agent typically has uncertain models for them.

states that it was executed from, e.g. if you make the decision to walk to your kitchen, the end result will be the same regardless of where you started from.

For subgoal options, the image operator can be replaced with the *effects distribution*: $\mathrm{Eff}(o) = \mathrm{Im}(o, Z), \forall Z(S)$, the resulting distribution over states after executing $o$ from any start distribution $Z(S)$. Planning with a set of subgoal options is simple because for each ordered pair of options $o_i$ and $o_j$, it is possible to compute and store $G(o_i, o_j)$, the probability that $o_j$ can be executed immediately after executing $o_i$: $G(o_i, o_j) = \int_S \mathrm{Pre}(o_j, s)\mathrm{Eff}(o_i)(s)\mathrm{d}s$.

We use the following two generalizations of subgoal options: *abstract subgoal options* model the more general case where executing an option leads to a subgoal for a subset of the state variables (called the *mask*), leaving the rest unchanged. For example, walking to the kitchen leaves the amount of gas in your car unchanged. More formally, the state vector can be partitioned into two parts $s = [a, b]$, such that executing $o$ leaves the agent in state $s' = [a, b']$, where $P(b')$ is independent of the distribution over starting states. The second generalization is the (abstract) *partitioned subgoal option*, which can be partitioned into a finite number of (abstract) subgoal options. For instance, an option for opening doors is not a subgoal option because there are many doors in the world, however it can be partitioned into a set of subgoal options, with one for every door.

The subgoal (and abstract subgoal) assumptions propose that the exact state from which option execution starts does not really affect the options that can be executed next. This is somewhat restrictive and often does not hold for options as given, but can hold for options once they have been partitioned. Additionally, the assumptions need only hold approximately in practice.

## 3 Online Active Symbol Acquisition

Previous approaches for learning symbolic models from data [11, 12] used random exploration. However, real world data from high-level skills is very expensive to collect, so it is important to use a more data-efficient approach. In this section, we introduce a new method for learning abstract models data-efficiently. Our approach maintains a distribution over symbolic models which is updated after every new observation. This distribution is used to choose the sequence of options that in expectation maximally reduces the amount of uncertainty in the posterior distribution over models. Our approach has two components: an active exploration algorithm which takes as input a distribution over symbolic models and returns the next option to execute, and an algorithm for quickly building a distribution over symbolic models from data. The second component is an improvement upon previous approaches in that it returns a distribution and is fast enough to be updated online, both of which we require.

### 3.1 Fast Construction of a Distribution over Symbolic Option Models

Now we show how to construct a more general model than $G$ that can be used for planning with abstract partitioned subgoal options. The advantages of our approach versus previous methods are that our algorithm is much faster, and the resulting model is Bayesian, both of which are necessary for the active exploration algorithm introduced in the next section.

Recall that the agent can collect observations of the forms $(s, o, s')$ upon executing option $o$ from $s$, and $(s, O(s))$ when entering a state $s$, where $O(s)$ is the set of available options in state $s$. Given a sequence of observations of this form, the first step of our approach is to find the *factors* [12],

partitions of state variables that always change together in the observed data. For example, consider a robot which has options for moving to the nearest table and picking up a glass on an adjacent table. Moving to a table changes the $x$ and $y$ coordinates of the robot without changing the joint angles of the robot's arms, while picking up a glass does the opposite. Thus, the $x$ and $y$ coordinates and the arm joint angles of the robot belong to different factors. Splitting the state space into factors reduces the number of potential masks (see end of Section 2.2) because we assume that if state variables $i$ and $j$ always change together in the observations, then this will always occur, e.g. we assume that moving to the table will never move the robot's arms.[1]

**Finding the Factors**  Compute the set of observed masks $M$ from the $(s, o, s')$ observations: each observation's mask is the subset of state variables that differ substantially between $s$ and $s'$. Since we work in continuous, stochastic domains, we must detect the difference between minor random noise (independent of the action) and a substantial change in a state variable caused by action execution. In principle this requires modeling action-independent and action-dependent differences, and distinguishing between them, but this is difficult to implement. Fortunately we have found that in practice allowing some noise and having a simple threshold is often effective, even in more noisy and complex domains. For each state variable $i$, let $M_i \subseteq M$ be the subset of the observed masks that contain $i$. Two state variables $i$ and $j$ belong to the same factor $f \in F$ if and only if $M_i = M_j$. Each factor $f$ is given by a set of state variables and thus corresponds to a subspace $S_f$. The factors are updated after every new observation.

Let $S^*$ be the set of states that the agent has observed and let $S_f^*$ be the projection of $S^*$ onto the subspace $S_f$ for some factor $f$, e.g. in the previous example there is a $S_f^*$ which consists of the set of observed robot $(x, y)$ coordinates. It is important to note that the agent's observations come only from executing partitioned abstract subgoal options. This means that $S_f^*$ consists only of abstract subgoals, because for each $s \in S^*$, $s_f$ was either unchanged from the previous state, or changed to another abstract subgoal. In the robot example, all $(x, y)$ observations must be adjacent to a table because the robot can only execute an option that terminates with it adjacent to a table or one that does not change its $(x, y)$ coordinates. Thus, the states in $S^*$ can be imagined as a collection of abstract subgoals for each of the factors. Our next step is to build a set of symbols for each factor to represent its abstract subgoals, which we do using unsupervised clustering.

**Finding the Symbols**  For each factor $f \in F$, we find the set of symbols $Z^f$ by clustering $S_f^*$. Let $Z^f(s_f)$ be the corresponding symbol for state $s$ and factor $f$. We then map the observed states $s \in S^*$ to their corresponding symbolic states $s^d = \{Z^f(s_f), \forall f \in F\}$, and the observations $(s, O(s))$ and $(s, o, s')$ to $(s^d, O(s))$ and $(s^d, o, s'^d)$, respectively.

In the robot example, the $(x, y)$ observations would be clustered around tables that the robot could travel to, so there would be a symbol corresponding to each table.

We want to build our models within the symbolic state space $S^d$. Thus we define the *symbolic precondition*, $Pre(o, s^d)$, which returns the probability that the agent can execute an option from some symbolic state, and the *symbolic effects distribution* for a subgoal option $o$, $Eff(o)$, maps to a subgoal distribution over symbolic states. For example, the robot's 'move to the nearest table' option maps the robot's current $(x, y)$ symbol to the one which corresponds to the nearest table.

The next step is to partition the options into abstract subgoal options (in the symbolic state space), e.g. we want to partition the 'move to the nearest table' option in the symbolic state space so that the symbolic states in each partition have the same nearest table.

**Partitioning the Options**  For each option $o$, we initialize the partitioning $P^o$ so that each symbolic state starts in its own partition. We use independent Bayesian sparse Dirichlet-categorical models [18] for the symbolic effects distribution of each option partition.[2] We then perform Bayesian Hierarchical Clustering [8] to merge partitions which have similar symbolic effects distributions.[3]

**Algorithm 1** Fast Construction of a Distribution over Symbolic Option Models

---
1: Find the set of observed masks $M$.
2: Find the factors $F$.
3: $\forall f \in F$, find the set of symbols $Z^f$.
4: Map the observed states $s \in S^*$ to symbolic states $s^d \in S^{*d}$.
5: Map the observations $(s, O(s))$ and $(s, o, s')$ to $(s^d, O(s))$ and $(s^d, o, s'^d)$.
6: $\forall o \in O$, initialize $P^o$ and perform Bayesian Hierarchical Clustering on it.
7: $\forall o \in O$, find $A^o$ and $F_*^o$.

---

There is a special case where the agent has observed that an option $o$ was available in some symbolic states $S_a^d$, but has yet to actually execute it from any $s^d \in S_a^d$. These are not included in the Bayesian Hierarchical Clustering, instead we have a special prior for the partition of $o$ that they belong to. After completing the merge step, the agent has a partitioning $P^o$ for each option $o$. Our prior is that with probability $q_o$,[4] each $s^d \in S_a^d$ belongs to the partition $p^o \in P^o$ which contains the symbolic states most similar to $s^d$, and with probability $1 - q_o$ each $s^d$ belongs to its own partition. To determine the partition which is most similar to some symbolic state, we first find $A^o$, the smallest subset of factors which can still be used to correctly classify $P^o$. We then map each $s^d \in S_a^d$ to the most similar partition by trying to match $s^d$ masked by $A^o$ with a masked symbolic state already in one of the partitions. If there is no match, $s^d$ is placed in its own partition.

Our final consideration is how to model the symbolic preconditions. The main concern is that many factors are often irrelevant for determining if some option can be executed. For example, whether or not you have keys in your pocket does not affect whether you can put on your shoe.

**Modeling the Symbolic Preconditions**  Given an option $o$ and subset of factors $F^o$, let $S_{F^o}^d$ be the symbolic state space projected onto $F^o$. We use independent Bayesian Beta-Bernoulli models for the symbolic precondition of $o$ in each masked symbolic state $s_{F^o}^d \in S_{F^o}^d$. For each option $o$, we use Bayesian model selection to find the the subset of factors $F_*^o$ which maximizes the likelihood of the symbolic precondition models.

The final result is a distribution over symbolic option models $H$, which consists of the combined sets of independent symbolic precondition models $\{Pre(o, s_{F_*^o}^d); \forall o \in O, \forall s_{F_*^o}^d \in S_{F_*^o}^d\}$ and independent symbolic effects distribution models $\{Eff(o, p^o); \forall o \in O, \forall p^o \in P^o\}$.

The complete procedure is given in Algorithm 1. A symbolic option model $h \sim H$ can be sampled by drawing parameters for each of the Bernoulli and categorical distributions from the corresponding Beta and sparse Dirichlet distributions, and drawing outcomes for each $q_o$. It is also possible to consider distributions over other parts of the model such as the symbolic state space and/or a more complicated one for the option partitionings, which we leave for future work.

## 3.2   Optimal Exploration

In the previous section we have shown how to efficiently compute a distribution over symbolic option models $H$. Recall that the ultimate purpose of $H$ is to compute the success probabilities of plans (see Section 2.2). Thus, the quality of $H$ is determined by the accuracy of its predicted plan success probabilities, and efficiently learning $H$ corresponds to selecting the sequence of observations which maximizes the expected accuracy of $H$. However, it is difficult to calculate the expected accuracy of $H$ over all possible plans, so we define a proxy measure to optimize which is intended to represent the amount of uncertainty in $H$. In this section, we introduce our proxy measure, followed by an algorithm for finding the exploration policy which optimizes it. The algorithm operates in an online manner, building $H$ from the data collected so far, using $H$ to select an option to execute, updating $H$ with the new observation, and so on.

First we define the *standard deviation* $\sigma_H$, the quantity we use to represent the amount of uncertainty in $H$. To define the standard deviation, we need to also define the distance and mean.

We define the *distance K* from $h_2 \in H$ to $h_1 \in H$, to be the sum of the Kullback-Leibler (KL) divergences[5] between their individual symbolic effect distributions plus the sum of the KL divergences between their individual symbolic precondition distributions:[6]

$$K(h_1, h_2) = \sum_{o \in O} [ \sum_{s_{F_*}^d \in S_{F_*}^d} D_{KL}(Pre^{h_1}(o, s_{F_*}^d) \| Pre^{h_2}(o, s_{F_*}^d))$$

$$+ \sum_{p^o \in P^o} D_{KL}(Eff^{h_1}(o, p^o) \| Eff^{h_2}(o, p^o))].$$

We define the *mean*, $\mathbb{E}[H]$, to be the symbolic option model such that each Bernoulli symbolic precondition and categorical symbolic effects distribution is equal to the mean of the corresponding Beta or sparse Dirichlet distribution:

$$\forall o \in O, \ \forall p^o \in P^o, \ Eff^{\mathbb{E}[H]}(o, p^o) = \mathbb{E}_{h \sim H}[Eff^h(o, p^o)],$$

$$\forall o \in O, \ \forall s_{F_*}^d \in S_{F_*}^d, \ Pre^{\mathbb{E}[H]}(o, s_{F_*}^d)) = \mathbb{E}_{h \sim H}[Pre^h(o, s_{F_*}^d))].$$

The standard deviation $\sigma_H$ is then simply: $\sigma_H = \mathbb{E}_{h \sim H}[K(h, \mathbb{E}[H])]$. This represents the expected amount of information which is lost if $\mathbb{E}[H]$ is used to approximate $H$.

Now we define the optimal exploration policy for the agent, which aims to maximize the expected reduction in $\sigma_H$ after $H$ is updated with new observations. Let $H(w)$ be the posterior distribution over symbolic models when $H$ is updated with symbolic observations $w$ (the partitioning is not updated, only the symbolic effects distribution and symbolic precondition models), and let $W(H, i, \pi)$ be the distribution over symbolic observations drawn from the posterior of $H$ if the agent follows policy $\pi$ for $i$ steps. We define the optimal exploration policy $\pi^*$ as:

$$\pi^* = \underset{\pi}{\operatorname{argmax}} \ \sigma_H - \mathbb{E}_{w \sim W(H, i, \pi)}[\sigma_{H(w)}].$$

For the convenience of our algorithm, we rewrite the second term by switching the order of the expectations: $\mathbb{E}_{w \sim W(H, i, \pi)}[\mathbb{E}_{h \sim H(w)}[K(h, \mathbb{E}[H(w)])]] = \mathbb{E}_{w \sim W(h, i, \pi)}[K(h, \mathbb{E}[H(w)])]$.

Note that the objective function is non-Markovian because $H$ is continuously updated with the agent's new observations, which changes $\sigma_H$. This means that $\pi^*$ is non-stationary, so Algorithm 2 approximates $\pi^*$ in an online manner using Monte-Carlo tree search (MCTS) [3] with the UCT tree policy [10]. $\pi_T$ is the combined tree and rollout policy for MCTS, given tree $T$.

There is a special case when the agent simulates the observation of a previously unobserved transition, which can occur under the sparse Dirichlet-categorical model. In this case, the amount of information gained is very large, and furthermore, the agent is likely to transition to a novel symbolic state. Rather than modeling the unexplored state space, instead, if an unobserved transition is encountered during an MCTS update, it immediately terminates with a large bonus to the score, a similar approach to that of the R-max algorithm [2]. The form of the bonus is $-zg$, where $g$ is the depth that the update terminated and $z$ is a constant. The bonus reflects the opportunity cost of not experiencing something novel as quickly as possible, and in practice it tends to dominate (as it should).

## 4   The Asteroids Domain

The Asteroids domain is shown in Figure 2a and was implemented using physics simulator pybox2d. The agent controls a ship by either applying a thrust in the direction it is facing or applying a torque in either direction. The goal of the agent is to be able to navigate the environment without colliding with any of the four "asteroids." The agent's starting location is next to asteroid 1. The agent is given the following 6 options (see Appendix A for additional details):

1. **move-counterclockwise** and **move-clockwise**: the ship moves from the current face it is adjacent to, to the midpoint of the face which is counterclockwise/clockwise on the same asteroid from the current face. Only available if the ship is at an asteroid.

**Algorithm 2** Optimal Exploration
---
**Input:** Number of remaining option executions $i$.
1: **while** $i \geq 0$ **do**
2:     Build $H$ from observations (Algorithm 1).
3:     Initialize tree $T$ for MCTS.
4:     **while** number updates < threshold **do**
5:         Sample a symbolic model $h \sim H$.
6:         Do an MCTS update of $T$ with dynamics given by $h$.
7:         Terminate current update if depth $g$ is $\geq i$, or unobserved transition is encountered.
8:         Store simulated observations $w \sim W(h, g, \pi_T)$.
9:         Score = $K(h, \mathbb{E}[H]) - K(h, \mathbb{E}[H(w)]) - zg$.
10:     **end while**
11:     **return** most visited child of root node.
12:     Execute corresponding option; Update observations; $i$--.
13: **end while**
---

2. **move-to-asteroid-1**, **move-to-asteroid-2**, **move-to-asteroid-3**, and **move-to-asteroid-4**: the ship moves to the midpoint of the closest face of asteroid 1-4 to which it has an unobstructed path. Only available if the ship is not already at the asteroid and an unobstructed path to some face exists.

Exploring with these options results in only one factor (for the entire state space), with symbols corresponding to each of the 35 asteroid faces as shown in Figure 2a.

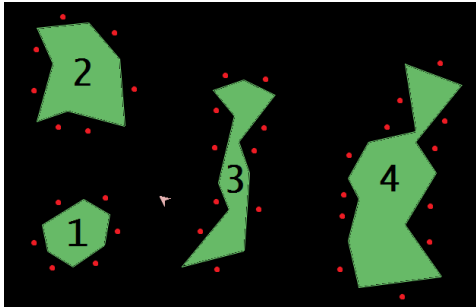

(a)

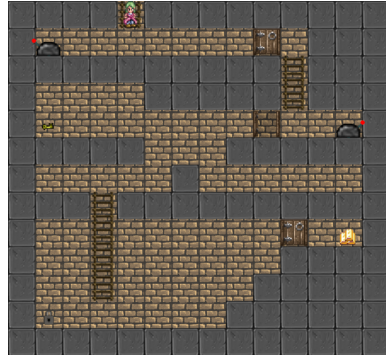

(b)

Figure 2: (a): The Asteroids Domain, and the 35 symbols which can be encountered while exploring with the provided options. (b): The Treasure Game domain. Although the game screen is drawn using large image tiles, sprite movement is at the pixel level.

**Results**  We tested the performance of three exploration algorithms: random, greedy, and our algorithm. For the greedy algorithm, the agent first computes the symbolic state space using steps 1-5 of Algorithm 1, and then chooses the option with the lowest execution count from its current symbolic state. The hyperparameter settings that we use for our algorithm are given in Appendix A.

Figures 3a, 3b, and 3c show the percentage of time that the agent spends on exploring asteroids 1, 3, and 4, respectively. The random and greedy policies have difficulty escaping asteroid 1, and are rarely able to reach asteroid 4. On the other hand, our algorithm allocates its time much more proportionally. Figure 4d shows the number of symbolic transitions that the agent has not observed (out of 115 possible).[7] As we discussed in Section 3, the number of unobserved symbolic transitions is a good representation of the amount of information that the models are missing from the environment.

Our algorithm significantly outperforms random and greedy exploration. Note that these results are using an uninformative prior and the performance of our algorithm could be significantly improved by

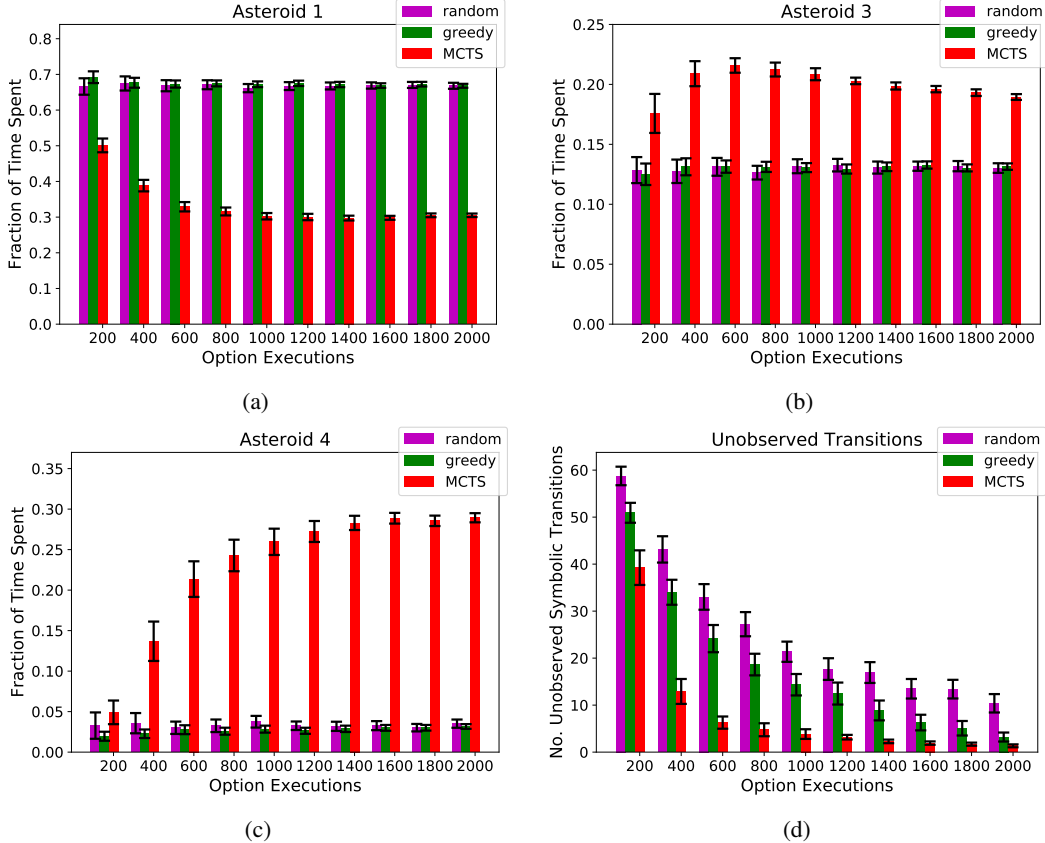

Figure 3: Simulation results for the Asteroids domain. Each bar represents the average of 100 runs. The error bars represent a 99% confidence interval for the mean. (a), (b), (c): The fraction of time that the agent spends on asteroids 1, 3, and 4, respectively. The greedy and random exploration policies spend significantly more time than our algorithm exploring asteroid 1 and significantly less time exploring asteroids 3 and 4. (d): The number of symbolic transitions that the agent has not observed (out of 115 possible). The greedy and random policies require 2-3 times as many option executions to match the performance of our algorithm.

starting with more information about the environment. To try to give additional intuition, in Appendix A we show heatmaps of the $(x, y)$ coordinates visited by each of the exploration algorithms.

## 5 The Treasure Game Domain

The Treasure Game [12], shown in Figure 2b, features an agent in a 2D, $528 \times 528$ pixel video-game like world, whose goal is to obtain treasure and return to its starting position on a ladder at the top of the screen. The 9-dimensional state space is given by the $x$ and $y$ positions of the agent, key, and treasure, the angles of the two handles, and the state of the lock.

The agent is given 9 options: go-left, go-right, up-ladder, down-ladder, jump-left, jump-right, down-right, down-left, and interact. See Appendix A for a more detailed description of the options and the environment dynamics. Given these options, the 7 factors with their corresponding number of symbols are: $player\text{-}x$, 10; $player\text{-}y$, 9; $handle1\text{-}angle$, 2; $handle2\text{-}angle$, 2; $key\text{-}x$ and $key\text{-}y$, 3; $bolt\text{-}locked$, 2; and $goldcoin\text{-}x$ and $goldcoin\text{-}y$, 2.

**Results** We tested the performance of the same three algorithms: random, greedy, and our algorithm. Figure 4a shows the fraction of time that the agent spends without having the key and with the lock still locked. Figures 4b and 4c show the number of times that the agent obtains the key and treasure, respectively. Figure 4d shows the number of unobserved symbolic transitions (out of 240 possible). Again, our algorithm performs significantly better than random and greedy exploration. The data

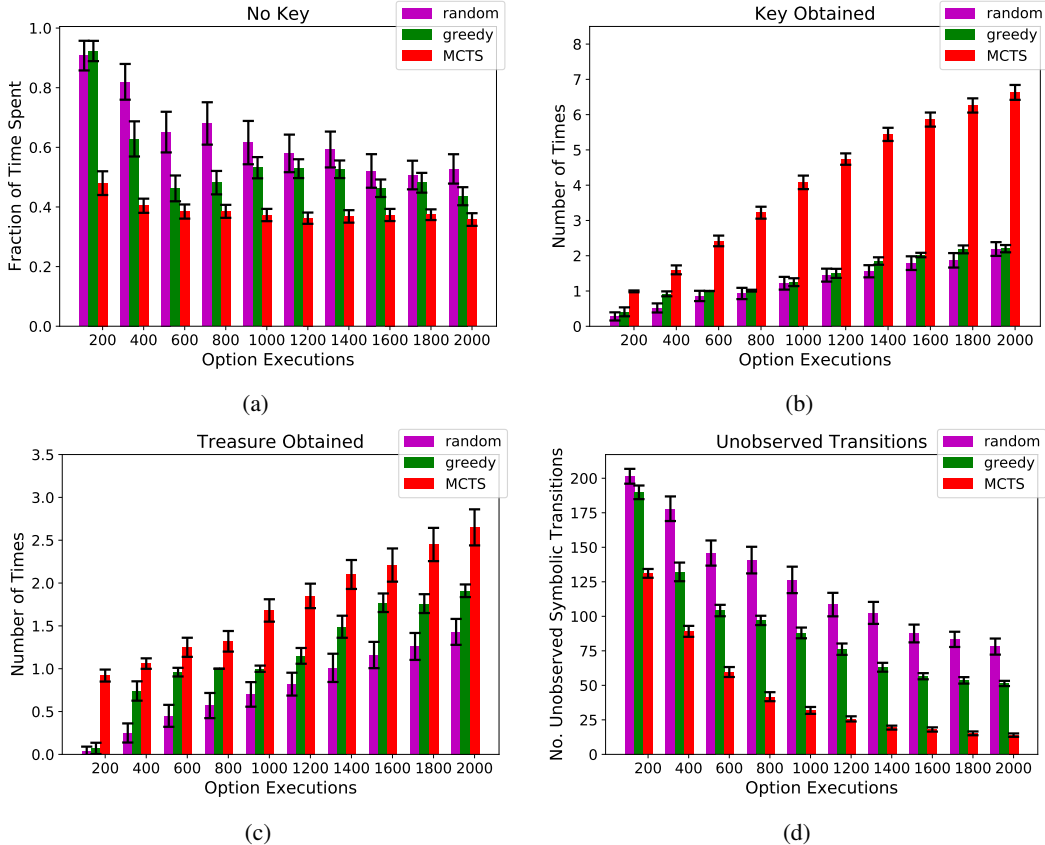

Figure 4: Simulation results for the Treasure Game domain. Each bar represents the average of 100 runs. The error bars represent a 99% confidence interval for the mean. (a): The fraction of time that the agent spends without having the key and with the lock still locked. The greedy and random exploration policies spend significantly more time than our algorithm exploring without the key and with the lock still locked. (b), (c): The average number of times that the agent obtains the key and treasure, respectively. Our algorithm obtains both the key and treasure significantly more frequently than the greedy and random exploration policies. There is a discrepancy between the number of times that our agent obtains the key and the treasure because there are more symbolic states where the agent can try the option that leads to a reset than where it can try the option that leads to obtaining the treasure. (d): The number of symbolic transitions that the agent has not observed (out of 240 possible). The greedy and random policies require 2-3 times as many option executions to match the performance of our algorithm.

from our algorithm has much better coverage, and thus leads to more accurate symbolic models. For instance in Figure 4c you can see that random and greedy exploration did not obtain the treasure after 200 executions; without that data the agent would not know that it should have a symbol that corresponds to possessing the treasure.

# 6  Conclusion

We have introduced a two-part algorithm for data-efficiently learning an abstract symbolic representation of an environment which is suitable for planning with high-level skills. The first part of the algorithm quickly generates an intermediate Bayesian symbolic model directly from data. The second part guides the agent's exploration towards areas of the environment that the model is uncertain about. This algorithm is useful when the cost of data collection is high, as is the case in most real world artificial intelligence applications. Our results show that the algorithm is significantly more data efficient than using more naive exploration policies.

# 7 Acknowledgements

This research was supported in part by the National Institutes of Health under award number R01MH109177. The U.S. Government is authorized to reproduce and distribute reprints for Governmental purposes notwithstanding any copyright notation thereon. The content is solely the responsibility of the authors and does not necessarily represent the official views of the National Institutes of Health.

## Footnotes

[1]The factors assumption is not strictly necessary as we can assign each state variable to its own factor. However, using this uncompressed representation can lead to an exponential increase in the size of the symbolic state space and a corresponding increase in the sample complexity of learning the symbolic models.

[2]We use sparse Dirichlet-categorical models because there are a combinatorial number of possible symbolic state transitions, but we expect that each partition has non-zero probability for only a small number of them.

[3]We use the closed form solutions for Dirichlet-multinomial models provided by the paper.

[4]This is a user specified parameter.

[5]The KL divergence has previously been used in other active exploration scenarios [16, 14].

[6]Similarly to other active exploration papers, we define the distance to depend only on the transition models and not the reward models.

[7]We used Algorithm 1 to build symbolic models from the data gathered by each exploration algorithms.

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
