[Supplementary Material · appendixNIPS.pdf]



Figure 1: Heatmaps of the $(x, y)$ coordinates visited by each exploration algorithm in the Asteroids domain. The plot was generated by the `hexbin` function in `matplotlib`. Our algorithm explores the state space much more uniformly than the random and greedy exploration algorithms.

## A   Environment Descriptions

### A.1   Asteroids Domain Cont.

The agent's 6 options are implemented using PD controllers for the torque and thrust. The options do not always work as intended, sometimes the ship will crash during the execution of an option, which resets the environment. If the ship tries to move from asteroid $a$ to asteroid $b > a$, it crashes with probability $0.5$. As designed, these options do not have the subgoal property because the outcome of executing each option is dependent on which face of which asteroid the option was executed from. However, they are partitioned subgoal options because their outcome is only dependent on which asteroid face they were executed from.

**Results Cont.**   Figure 5 shows heatmaps of the $(x, y)$ coordinates visited by each exploration algorithm in the Asteroids domain. Our algorithm explores the state space much more uniformly than the random and greedy exploration algorithms.

### A.2   Treasure Game Domain Cont.

The low-level actions available to the agent are move up, down, left, and right, jump, and interact. The 4 movement actions move the agent between 2 and 4 pixels uniformly at random in the appropriate direction. There are three doors which may block the path of the agent. The top two doors are oppositely open and closed; flipping one of the two handles switches their status. The bottom door which guards the treasure can be opened by the agent obtaining the key and using it on the lock. The interact action is available when the agent is standing in front of a handle, or when it possesses the key and is standing in front of the lock. In the first case, executing the interact action flips the handle's position with probability $0.8$, and in the second case, the lock is unlocked and the agent loses the key. Whenever the agent has possession of the key and/or the treasure, they are displayed in

the lower-right corner of the screen. The agent returning to the top ladder resets the environment. The agent's 9 options are implemented using simple control loops:

1. **go-right** and **go-left**: the agent moves continuously right/left until it reaches a wall, edge, object it can interact with, or ladder. Only available when the agent's way is not directly blocked.

2. **up-ladder** and **down-ladder**: the agent ascends/descends a ladder. Only available when the agent is directly below/above a ladder.

3. **down-left** and **down-right**: the agent falls off an edge onto the nearest solid cell on its left/right. Only available when they would succeed.

4. **jump-left** and **jump-right**: the agent jumps and moves left/right for about 48 pixels. Only available when the area above the agent's head, and above its head and to the left/right, are clear.

5. **interact**: same as the low-level interact action.

These options, like the low-level actions they are composed of, all have at least a small amount of stochasticity in their outcomes. Additionally, when the agent executes one of the jump options to reach a faraway ledge, for instance when it is trying to get the key, it succeeds with probability 0.53, and misses the ledge and lands directly below with probability 0.47. These are abstract partitioned subgoal options.

### A.3 Hyperparameter Settings

In each run the agent had access to the exact number of option executions it had to explore with. For MCTS, we used the UCT tree policy with $c = 2$, a random rollout policy, and performed 1000 updates. Also, during UCT option selection, we normalized a node's score using the highest and lowest scores seen so far. For the sparse Dirichlet-Multinomial models, we used hyperparameter $\alpha = 0.5$ and the prior probability over the size of the support was given by a geometric distribution with parameter 0.5. For the state clustering (step 3 of Algorithm 1), we used the DBSCAN algorithm **?** implemented in scikit-learn **?** with parameters $min\text{-}samples = 1$, and $\epsilon = 1.5$ for the Asteroids domain and $\epsilon = 0.05$ for the Treasure Game domain. For Algorithm 2, we set $z = 10$. For all options $o$, we set $q_o = 0.3$.