[Reviews · NeurIPS 2017]

Reviewer 1



The paper proposes an active exploration scheme for more efficiently building a symbolic representation of the world in which the agent operates. This builds on earlier work that introduces a rich symbolic language for representing decision making problems in continuous state spaces. The current work proposes a two phase strategy - the first phase builds a Bayesian model for the options based on data gathered through execution of the options. The second phase uses of a MCTS based exploration strategy that drives the exploration based on the goodness of prediction of the occurrence of the symbols, given a set of options. They demonstrate the utility of the approach on two domains - one inspired by asteroid game and the other a treasure hunting game. The line of inquiry of this paper is very promising and quite needed. A truly intelligent agent should be able to actively seek experience that would better enable it to "understand" the domain. The paper is somewhat well written. The basic setting of the work is well explained, but it is a little hard to understand the framework proposed in this work, since the writing in sections 3.1 and 3.2 is a bit dense and hard to follow. It is not entirely clear what is the specific feature that makes the approach data efficient. The comparisons are against random and greedy exploration. Given that the greedy algorithm is a count based exploration strategy, it is surprising that it does so poorly. Is there some explanation for this? It would be nice to see some interpretation of the nature of the models learned when the active exploration strategy is used. One nitpick: If one ignores the duration of option completion, then what we have is a MDP and no longer a SMDP.

Reviewer 2



This is a very interesting paper, with multiple complementary ideas. It advocates model-based active exploration (model learning + seeking regions of uncertainty). Instead of doing this in raw state space, it proposes a method for abstracting states to symbols based on factoring and clustering the state space. The exploration is then done by MCTS-planning in a (sampled) symbolic model. The task setup evaluates pure exploration (ignoring all rewards) on a two different domains. This approach to unsupervised hierarchical reinforcement learning is novel and ambitious, the paper is clear and well-written. The proposed method may be somewhat brittle in its current form, and it is unclear to what problem complexity it can scale, but that can be resolved by future work. My main recommendation is to add a thorough discussion of its weaknesses and limitations. Other comments: * Section 2.1: maybe it’s not necessary to introduce discounts and rewards at all, given that neither are used in the paper? * Section 3.1: the method for finding the factors seems very brittle, and to rely on disentangled feature representations that are not noisy. Please discuss these limitations, and maybe hint at how factors could be found if the observations were a noisy sensory stream like vision. * Line 192: freezing the partitioning in the first iteration seems like a risky choice that makes strong assumptions about the coverage of the initial data. At least discuss the limitations of this. * Section 4: there is a mismatch between these options and the desired properties discussed in section 2.2: in particular, the proposed options are not “subgoal options” because their distribution over termination states strongly depends on the start states? Same for the Treasure Game. * Line 218: explicitly define what the “greedy” baseline is. * Figure 4: Comparing the greedy results between (b) and (c), it appears that whenever a key is obtained, the treasure is almost always found too, contrasting with the MCTS version that explores a lot of key-but-no-treasure states. Can you explain this?

Reviewer 3



Key question asked in the paper: Can an agent actively explore to build a probabilistic symbolic model of the environment from a given set of option and state abstractions? And does this help make future exploration easy via MCTS? This paper proposes a knowledge representation framework for addressing these questions. Proposal: A known continuous state space and discrete set of options (restricted to subgoal options) is given. An options model is treated a semi-MDP process. A plan is defined as a sequence of options. The idea of a probabilistic symbol is invoked from earlier work to refer to a distribution over infinitely many continuous low-level states. The idea of state masks is introduced to find independent factors of variations. Then each abstract subgoal option is defined as a policy that leads to a subgoal for the masked states, for e.g. opening a door. But since there could be many doors in the environment, the idea of a partitioned abstract subgoal option is proposed to bind together subgoal options for each instance of a door. The agent then uses these partitioned abstract subgoal options during exploration. Optimal exploration then is defined as finding a policy via MCTS that leads to greatest reduction of uncertainty over distributions over proposed options symbolic model. Feedback: - There are many ideas proposed in this paper. This makes it hard to clearly communicate the contributions. This work builds on previous work from Konidaris et al. Is the novel contribution in the MCTS algorithm? The contributions should be made more explicit for unfamiliar readers. - It is difficult to keep track of what's going on in sec 3. I would additionally make alg1 more self-inclusive and use one of the working examples (asteroid or maze) to explain things. - Discuss assumption of restriction to subgoal option. What kind of behaviors does this framework not permit? discuss this in the paper - Another restrictive assumption: "We use factors to reduce the number of potential masks, i.e. we assume that if state variables i and j always change together in the observations, then this will always occur. An example of a factor could be the (x, y, z) position of your keys, because they are almost never moved along only one axis". What are the implications of this assumptions on the possible range of behaviors? - The experimental setup needs to be developed more to answer, visualize and discuss a few interesting questions: (1) how does the symbol model change with active exploration? (2) visualize the space of policies at various intermediate epochs. Without this it is hard to get an intuition for the ideas and ways to expand them in the future. - I believe this is a very important line of work. However, clarity and articulation in the writing/experiments remains my main concern for a clear accept. Since some of the abstractions are fairly new, the authors also need to clearly discuss the implication of all their underlying assumptions.